# Evaluating the use of large language models to provide clinical recommendations in the Emergency Department

Christopher Y. K. Williams [1] ✉, Brenda Y. Miao [1], Aaron E. Kornblith [1,2] & Atul J. Butte [1]

The release of GPT-4 and other large language models (LLMs) has the potential to transform healthcare. However, existing research evaluating LLM performance on real-world clinical notes is limited. Here, we conduct a highly-powered study to determine whether LLMs can provide clinical recommendations for three tasks (admission status, radiological investigation(s) request status, and antibiotic prescription status) using clinical notes from the Emergency Department. We randomly selected 10,000 Emergency Department visits to evaluate the accuracy of zero-shot, GPT-3.5-turbo- and GPT-4-turbo-generated clinical recommendations across four different prompting strategies. We found that both GPT-4-turbo and GPT-3.5-turbo performed poorly compared to a resident physician, with accuracy scores 8% and 24%, respectively, lower than physician on average. Both LLMs tended to be overly cautious in its recommendations, with high sensitivity at the cost of specificity. Our findings demonstrate that, while early evaluations of the clinical use of LLMs are promising, LLM performance must be significantly improved before their deployment as decision support systems for clinical recommendations and other complex tasks.

Since its November 2022 launch, the Chat Generative Pre-Trained Transformer (ChatGPT; GPT-3.5-turbo) has captured widespread public attention, with media reports suggesting over 100 million monthly active users just 2 months after launch[1]. Along with its successor, GPT-4, these large language models (LLMs) use a chat-based interface to respond to complex queries and solve problems[2,3]. Although trained as general-purpose models, researchers have begun evaluating the performance of GPT-3.5-turbo and GPT-4 on clinically relevant tasks. For instance, GPT-3.5-turbo was found to provide largely appropriate responses when asked to give simple cardiovascular disease prevention recommendations[4]. Meanwhile, GPT-3.5-turbo responses to patients' health questions on a public social media forum were both preferred, and rated as having higher empathy, compared to physician responses[5].

While there are a growing number of studies that explore the uses of the GPT models across a range of clinical tasks, the majority do not use real-world clinical notes. They instead apply these models to answer questions from medical examinations such as the USMLE, solve publicly available clinical diagnostic challenges such as the *New England Journal of Medicine* (NEJM) clinicopathologic conferences, or evaluate performance on existing clinical benchmarks[3,6–9]. This is due to the challenges associated with disclosing protected health information (PHI) with LLM providers such as OpenAI in a Health Insurance Portability and Accountability Act (HIPAA) compliant manner, where business associate agreements must be in place to allow secure processing of PHI content[10]. This is a notable hurdle given the inherent differences between curated medical datasets, such as the USMLE question bank, and real-world clinical notes. In addition, this issue is

[1]Bakar Computational Health Sciences Institute, University of California, San Francisco, San Francisco, CA, USA. [2]Department of Emergency Medicine, University of California, San Francisco, CA, USA. ✉e-mail: cykw2@doctors.org.uk

particularly problematic when you consider that the GPT models have likely been trained on data obtained from open sources on the Internet and, therefore, their evaluation of existing publicly available benchmarks or tasks may be confounded by data leakage[11].

As the availability and accessibility of these models increase, it is now critically important to better understand the potential uses and limitations of LLMs applied to actual clinical notes. In our previous work, we showed that GPT-4 could accurately identify the higher acuity patient in pairs of Emergency Department patients when provided only the clinical histories[12]. This performance was seen despite a lack of additional training or fine-tuning, known as zero-shot learning[13]. Elsewhere, Kanjee and colleagues evaluated the diagnostic ability of GPT-4 across 70 cases from the NEJM clinicopathologic conferences, obtaining a correct diagnosis in its differential in 64% of cases and as its top diagnosis in 39%[7]. However, the ability of these general-purpose LLMs to assimilate clinical information from de-identified clinical notes and return clinical recommendations is still unclear.

In this study, we sought to evaluate the zero-shot performance of GPT-3.5-turbo and GPT-4-turbo when prompted to provide clinical recommendations for patients evaluated in the Emergency Department. We focus on three recommendations in particular: (1) Should the patient be admitted to hospital; (2) Should the patient have radiological investigations requested; and (3) Should the patient receive antibiotics? These are important considerations for Emergency Medicine providers in determining clinical direction, managing staffing and resources, and designating bed utilisation. We first evaluate performance on balanced (i.e., equal numbers of positive and negative outcomes) datasets to examine the sensitivity and specificity of GPT recommendations before determining overall model accuracy on an unbalanced dataset that reflects real-world distributions of patients presenting to the Emergency Department.

## Results

From a total of 251,401 adult Emergency Department visits, we first created balanced samples of 10,000 ED visits for each of the three tasks (Fig. 1). Using only the information provided in the *Presenting History* and *Physical Examination* sections of patients' first ED physician note, we queried GPT-3.5-turbo and GPT-4-turbo to determine whether (1) the patient should be admitted to hospital, (2) the patient requires radiological investigation(s), and (3) the patient requires antibiotics, comparing the output to the ground-truth outcome extracted from the electronic health record.

Across all three clinical recommendation tasks, overall GPT-3.5-turbo performance was poor (Table 1a). The initial prompt of 'Please return whether the patient should be admitted to hospital/requires radiological investigation/requires antibiotics' (Prompt A, see Supplementary Information) led to high sensitivity and low specificity performance. For this prompt, GPT-3.5-turbo recommendations had a high true positive rate but a similarly high false positive rate, with GPT-3.5-turbo recommending admission/radiological investigation/antibiotic prescription for the majority of cases. Altering the prompt to 'only suggest … if absolutely required' (Prompt B) only marginally improved specificity. The greatest performance was achieved by removing restrictions on the verbosity of GPT-3.5-turbo response (Prompt C) and adding the 'Let's think step by step' chain-of-thought prompting (Prompt D). These prompts generated the highest specificity in GPT-3.5-turbo recommendations with limited effect on sensitivity. On the evaluation of GPT-4-turbo performance, there were similar sensitivity and specificity scores for the *Admission status* task compared to the GPT-3.5-turbo model. In contrast, across both *Radiology investigation request status* and *Antibiotic prescription status* tasks, GPT-4-turbo demonstrated marked improvement in specificity at the expense of sensitivity (Table 1b).

To compare this performance with that of a resident physician, for each of the three tasks, we took a balanced n = 200 subsample (Fig. 1)

for manual annotation and compared performance between physician and machine across the four prompt iterations (Table 2). Notably, physician sensitivity was below that of GPT-3.5-turbo responses, whereas specificity was significantly higher (Table 2a). There were similar findings when comparing GPT-4-turbo performance to physician, except for the *Antibiotic prescription status* task where GPT-4-turbo specificity surpassed that of the physician, but had worse sensitivity (Table 2b).

We next sought to test the performance of the LLMs in a more representative setting using an unbalanced, n = 1000 sample of ED visits that reflects the real-world distribution of admission, radiological investigation, and antibiotic prescription rates at our institution (Table 3). We found that the accuracy of resident physician recommendations, when evaluated against the ground-truth outcomes extracted from the electronic health record, was significantly higher than GPT-3.5-turbo recommendations: 0.83 for physician vs [range of accuracy scores across Prompts A-D: 0.29–0.53], 0.79 vs [range of accuracy scores across Prompts A-D: 0.68–0.71] and 0.78 vs [range of accuracy scores across Prompts A-D: 0.35–0.43] for admission, radiological investigation, and antibiotic prescription tasks, respectively (Fig. 2; Table 3a). GPT-4-turbo accuracy was higher than that of its predecessor for every task across each of the prompts used (Fig. 2; Table 3b). However, GPT-4-turbo performance remained inferior to the physician for both *Admission status* ([range of accuracy scores across Prompts A-D: 0.43–0.58] vs physician accuracy of 0.83) and *Radiological investigation(s) request status* ([accuracy = 0.74 across Prompts A-D] vs physician accuracy of 0.79) tasks. For the *Antibiotic prescription status* task, GPT-4-turbo performance surpassed the physician (GPT-4-turbo accuracy of 0.83 vs physician accuracy of 0.78).

Lastly, in our sensitivity analyses conducted on a balanced, n = 200 subsample for each task, results were largely similar regardless of the written order of labels in the original prompt (e.g., '0: Patient should not be admitted to hospital. 1: Patient should be admitted to hospital.' Vs '1: Patient should be admitted to hospital. 0: Patient should not be admitted to hospital.') (Tables S4 and S5).

## Discussion

This study represents an early, highly-powered evaluation of the potential uses and limitations of LLMs for generating clinical recommendations based on real-world clinical text. Across three different clinical recommendation tasks, we found that GPT-3.5-turbo performed poorly, with high sensitivity but low specificity across tasks. GPT-4-turbo performed better than its predecessor, especially when predicting the need for antibiotics for a patient in the Emergency Department. Model performance was marginally improved with iterations of prompt engineering, including the addition of zero-shot chain-of-thought prompting[14]. On the evaluation of an unbalanced sample reflective of the real-world distribution of clinical recommendations, the overall performance of both GPT-4-turbo and GPT-3.5-turbo was significantly worse than that of a resident physician, with 8% and 24% lower accuracy, respectively, when averaged across tasks. However, there were notable differences in GPT-4-turbo performance across tasks, with 25% lower accuracy compared to physicians for *Admission status* ('Should the patient be admitted to hospital?') and 4% lower accuracy for *Radiological investigation(s) request status* ('Should the patient have radiological investigations requested?'), compared with 5% greater accuracy for *Antibiotic prescription status* ('Should the patient receive antibiotics?').

Our results suggest that LLMs are overly cautious in their clinical recommendations−both models exhibit a tendency to recommend intervention and this leads to a notable number of false positive suggestions. Such a finding is problematic given the need to both prioritise hospital resource availability and reduce overall healthcare costs[15,16]. This is also true at the patient level, where there is an

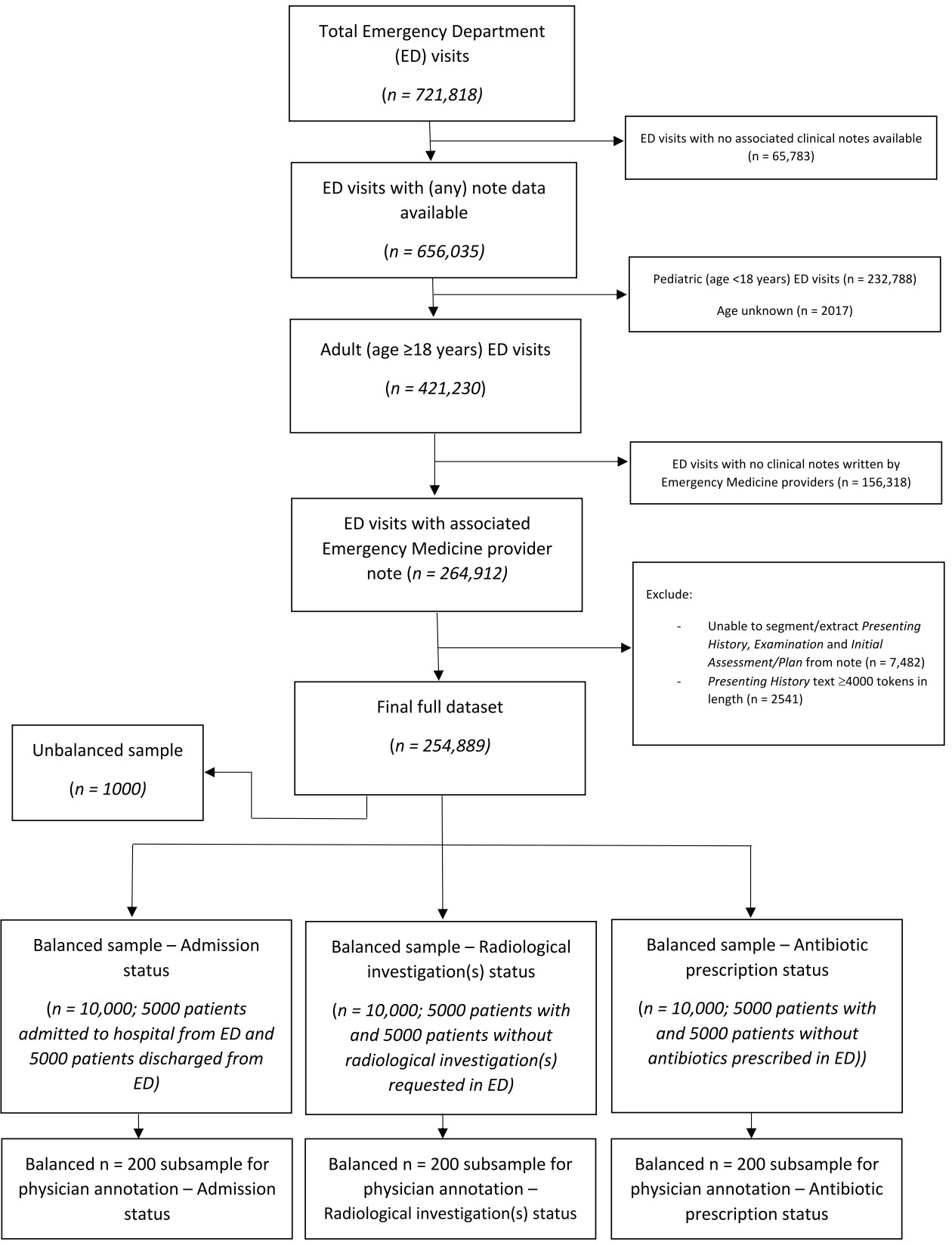

**Fig. 1 | Patient flowchart.** Flowchart of included Emergency Department visits and construction of both balanced (*n* = 10,000 samples) and unbalanced (*n* = 1000 samples reflecting the real-world distribution of patients presenting to the Emergency Department) datasets for the following outcomes: (1) admission status, (2) radiological investigation(s) status, and (3) antibiotic prescription status.

**Table 1 | (a) GPT-3.5-turbo performance and (b) GPT-4-turbo performance across four iterations of prompt engineering (Prompt A-D) evaluated on a balanced n = 10,000 sample for three clinical recommendation tasks: (1) should the patient be admitted to hospital; (2) does the patient require radiological investigation; and (3) does the patient require antibiotics**

| | Task | | True positives, n (%) | False positives, n (%) | True negatives, n (%) | False negatives, n (%) | Sensitivity (95% CIª) | Specificity (95% CIª) |
|---|---|---|---|---|---|---|---|---|
| a) GPT-3.5-turbo | 1) Admission status | Prompt A | 4994 (49.9) | 4639 (46.4) | 361 (3.6) | 6 (0.1) | **1.00 (1.00–1.00)** | 0.07 (0.07–0.08) |
| | | Prompt B | 4904 (49.0) | 3527 (35.3) | 1473 (14.7) | 96 (1) | 0.98 (0.98–0.98) | 0.29 (0.28–0.31) |
| | | Prompt C | 4683 (46.8) | 3255 (32.6) | 1745 (17.5) | 317 (3.2) | 0.94 (0.93–0.94) | 0.35 (0.34–0.36) |
| | | Prompt D | 4617 (46.2) | 3165 (31.7) | 1835 (18.4) | 383 (3.8) | 0.92 (0.92–0.93) | **0.37 (0.35–0.38)** |
| | 2) Radiological investigation(s) request status | Prompt A | 4922 (49.2) | 4361 (43.6) | 639 (6.4) | 78 (0.8) | **0.98 (0.98–0.99)** | 0.13 (0.12–0.14) |
| | | Prompt B | 4805 (48.1) | 3906 (39.1) | 1094 (10.9) | 195 (2) | 0.96 (0.96–0.97) | 0.22 (0.21–0.23) |
| | | Prompt C | 4792 (47.9) | 3855 (38.6) | 1145 (11.5) | 208 (2.1) | 0.96 (0.95–0.96) | **0.23 (0.22–0.24)** |
| | | Prompt D | 4819 (48.2) | 3991 (39.9) | 1009 (10.1) | 181 (1.8) | 0.96 (0.96–0.97) | 0.20 (0.19–0.21) |
| | 3) Antibiotic prescription status | Prompt A | 4812 (48.1) | 3955 (39.6) | 1045 (10.5) | 188 (1.9) | **0.96 (0.96–0.97)** | 0.21 (0.20–0.22) |
| | | Prompt B | 4690 (46.9) | 3687 (36.9) | 1313 (13.1) | 310 (3.1) | 0.94 (0.93–0.95) | 0.26 (0.25–0.28) |
| | | Prompt C | 4658 (46.6) | 3639 (36.4) | 1361 (13.6) | 342 (3.4) | 0.93 (0.92–0.94) | 0.27 (0.26–0.29) |
| | | Prompt D | 4544 (45.4) | 3379 (33.8) | 1621 (16.2) | 456 (4.6) | 0.91 (0.90–0.92) | **0.32 (0.31–0.34)** |
| b) GPT-4-turbo | 1) Admission status | Prompt A | 4986 (49.9) | 3778 (37.8) | 1222 (12.2) | 14 (0.1) | **1.00 (1.00–1.00)** | 0.24 (0.23–0.26) |
| | | Prompt B | 4908 (49.1) | 2982 (29.8) | 2018 (20.2) | 92 (0.9) | 0.98 (0.98–0.99) | 0.40 (0.39–0.42) |
| | | Prompt C | 4889 (48.9) | 2925 (29.3) | 2075 (20.8) | 111 (1.1) | 0.98 (0.97–0.98) | **0.42 (0.40–0.43)** |
| | | Prompt D | 4925 (49.3) | 3147 (31.5) | 1853 (18.5) | 75 (0.8) | 0.98 (0.98–0.99) | 0.37 (0.36–0.38) |
| | 2) Radiological investigation(s) request status | Prompt A | 4508 (45.1) | 2707 (27.1) | 2293 (22.9) | 492 (4.9) | **0.90 (0.89–0.91)** | 0.46 (0.44–0.47) |
| | | Prompt B | 4006 (40.1) | 1867 (18.7) | 3133 (31.3) | 994 (9.9) | 0.8 (0.79–0.81) | 0.63 (0.61–0.64) |
| | | Prompt C | 3796 (38) | 1653 (16.5) | 3347 (33.5) | 1204 (12) | 0.76 (0.75–0.77) | **0.67 (0.66–0.68)** |
| | | Prompt D | 4107 (41.1) | 2016 (20.2) | 2984 (29.8) | 893 (8.9) | 0.82 (0.81–0.83) | 0.60 (0.58–0.61) |
| | 3) Antibiotic prescription status | Prompt A | 3149 (31.5) | 675 (6.8) | 4325 (43.3) | 1851 (18.5) | **0.63 (0.62–0.64)** | 0.86 (0.86–0.87) |
| | | Prompt B | 2711 (27.1) | 482 (4.8) | 4518 (45.2) | 2289 (22.9) | 0.54 (0.53–0.56) | 0.90 (0.90–0.91) |
| | | Prompt C | 2505 (25.1) | 428 (4.3) | 4572 (45.7) | 2495 (25) | 0.50 (0.49–0.52) | **0.91 (0.91–0.92)** |
| | | Prompt D | 2584 (25.8) | 452 (4.5) | 4548 (45.5) | 2416 (24.2) | 0.52 (0.50–0.53) | 0.91 (0.90–0.92) |

For each task and metric, the best performing scores are highlighted in bold.
ª95% confidence intervals (CIs) calculated by bootstrapping.

increasing appreciation that excessive investigation and/or treatment may cause patients harm[16]. It is unclear, however, what is the best balance of sensitivity/specificity to strive for amongst clinical LLMs—it is likely that this balance will differ based on the particular task. The increase in LLM specificity, at the expense of sensitivity, across our iterations of prompt engineering suggests that improvements could be made bespoke to the task, though the extent to which prompt engineering alone may improve performance is unclear.

Across all three tasks, overall performance remained notably below that of a human physician. This may reflect the inherent complexity of clinical decision making, where clinical recommendations may be influenced not only by the patient's intrinsic clinical status, but also by patient preference, current resource availability and other external factors, such as social determinants of health and resources available at home.

Before LLMs can be integrated into the clinical environment, it is important to fully understand both their capabilities and limitations. Otherwise, there is a risk of unintended harmful consequences, especially if models have been deployed at scale[17,18]. Current research deploying LLMs, particularly the current state-of-the-art GPT models, on real-world clinical text is limited. Recent work from our group has demonstrated the accurate performance of GPT-4 in assessing patient clinical acuity in the Emergency Department, identifying reasons for contraceptive switching, and extracting detailed oncologic history and treatment plans from medical oncology notes[12,19,20]. Elsewhere, GPT-3.5-turbo has been used to convert radiology reports into plain language, to classify whether statements of clinical recommendations in

scientific literature constitute health advice, and to accurately classify five diseases from discharge summaries in the MIMIC-III dataset[21–23]. Much of the current literature focuses on the strengths of LLMs such as GPT-3.5-turbo and GPT-4[3,9,12,19,20]. However, it is equally important to identify areas of medicine in which LLMs do not perform well. For example, in one evaluation of GPT-4's ability to diagnose dementia from a set of structured features, GPT-4 did not surpass the performance of traditional AI tools, while fewer than 20% of GPT-3.5-turbo and GPT-4 responses submitted to a clinical informatics consult service were found to be concordant with existing reports[24,25]. Similarly, GPT-4 has been shown to produce largely accurate discharge summaries in the Emergency Department, but these summaries were liable to hallucination and omission of clinically relevant information[26]. While early signs of the utility of LLMs in medicine are promising, our findings suggest that there remains significant room for improvement, especially in more challenging tasks such as complex clinical decision-making.

This study has several limitations. First, it is possible that, for each task, not all the information which led to the real-life clinical recommendation extracted from the electronic health record was present in the *Presenting History* and *Physical Examination* sections of the ED physician note. For instance, radiological investigations requested following the Emergency Medicine physician review may lead to unexpected and/or incidental findings which were not detected during the initial review and may warrant admission or antibiotic prescription. However, even with this limitation, physician classification performance remained at a very respectable 78–83% accuracy across the

**Table 2 | Comparison of physician performance with (a) GPT-3.5-turbo performance and (b) GPT-4-turbo performance across four iterations of prompt engineering [Prompt A-D] evaluated on a balanced n = 200 subsample for three clinical recommendation tasks: (1) should the patient be admitted to hospital; (2) does the patient require radiological investigation; and (3) does the patient require antibiotics**

| | Task | | True positives, n (%) | False positives, n (%) | True negatives, n (%) | False Negatives, n (%) | Sensitivity (95% CI[a]) | Specificity (95% CI[a]) |
|---|---|---|---|---|---|---|---|---|
| a) GPT-3.5-turbo | 1) Admission status | *Physician* | *73 (36.5)* | *26 (13)* | *74 (37)* | *27 (13.5)* | *0.73 (0.64–0.81)* | **0.74 (0.66–0.82)** |
| | | Prompt A | 100 (50) | 93 (46.5) | 7 (3.5) | 0 (0) | **1 (1–1)** | 0.07 (0.03–0.13) |
| | | Prompt B | 98 (49) | 67 (33.5) | 33 (16.5) | 2 (1) | 0.98 (0.95–1.00) | 0.33 (0.24–0.43) |
| | | Prompt C | 95 (47.5) | 61 (30.5) | 39 (19.5) | 5 (2.5) | 0.95 (0.90–0.99) | 0.39 (0.30–0.49) |
| | | Prompt D | 93 (46.5) | 60 (30) | 40 (20) | 7 (3.5) | 0.93 (0.87–0.97) | 0.40 (0.31–0.50) |
| | 2) Radiological investigation(s) request status | *Physician* | *76 (38)* | *21 (10.5)* | *79 (39.5)* | *24 (12)* | *0.76 (0.67–0.84)* | **0.79 (0.70–0.87)** |
| | | Prompt A | 96 (48) | 91 (45.5) | 9 (4.5) | 4 (2) | **0.96 (0.92–0.99)** | 0.09 (0.04–0.15) |
| | | Prompt B | 93 (46.5) | 83 (41.5) | 17 (8.5) | 7 (3.5) | 0.93 (0.87–0.98) | 0.17 (0.09–0.24) |
| | | Prompt C | 95 (47.5) | 83 (41.5) | 17 (8.5) | 5 (2.5) | 0.95 (0.90–0.99) | 0.17 (0.10–0.24) |
| | | Prompt D | 95 (47.5) | 84 (42) | 16 (8) | 5 (2.5) | 0.95 (0.90–0.99) | 0.16 (0.09–0.24) |
| | 3) Antibiotic prescription status | *Physician* | *64 (32)* | *22 (11)* | *78 (39)* | *36 (18)* | *0.64 (0.55–0.73)* | **0.78 (0.70–0.86)** |
| | | Prompt A | 93 (46.5) | 74 (37) | 26 (13) | 7 (3.5) | **0.93 (0.88–0.97)** | 0.26 (0.18–0.35) |
| | | Prompt B | 91 (45.5) | 71 (35.5) | 29 (14.5) | 9 (4.5) | 0.91 (0.85–0.96) | 0.29 (0.20–0.39) |
| | | Prompt C | 92 (46) | 68 (34) | 32 (16) | 8 (4) | 0.92 (0.87–0.97) | 0.32 (0.23–0.42) |
| | | Prompt D | 89 (44.5) | 63 (31.5) | 37 (18.5) | 11 (5.5) | 0.89 (0.83–0.95) | 0.37 (0.27–0.47) |
| b) GPT-4-turbo | 1) Admission status | *Physician* | *73 (36.5)* | *26 (13)* | *74 (37)* | *27 (13.5)* | *0.73 (0.64–0.81)* | **0.74 (0.66–0.82)** |
| | | Prompt A | 100 (50) | 78 (39) | 22 (11) | 0 (0) | **1 (1–1)** | 0.22 (0.14–0.31) |
| | | Prompt B | 96 (48) | 59 (29.5) | 41 (20.5) | 4 (2) | 0.96 (0.92–0.99) | 0.41 (0.32–0.51) |
| | | Prompt C | 94 (47) | 56 (28) | 44 (22) | 6 (3) | 0.94 (0.89–0.99) | 0.44 (0.34–0.54) |
| | | Prompt D | 99 (49.5) | 67 (33.5) | 33 (16.5) | 1 (0.5) | 0.99 (0.97–1) | 0.33 (0.24–0.42) |
| | 2) Radiological investigation(s) request status | *Physician* | *76 (38)* | *21 (10.5)* | *79 (39.5)* | *24 (12)* | *0.76 (0.67–0.84)* | **0.79 (0.70–0.87)** |
| | | Prompt A | 88 (44) | 61 (30.5) | 39 (19.5) | 12 (6) | **0.88 (0.82–0.94)** | 0.39 (0.29–0.48) |
| | | Prompt B | 79 (39.5) | 37 (18.5) | 63 (31.5) | 21 (10.5) | 0.79 (0.71–0.86) | 0.63 (0.53–0.72) |
| | | Prompt C | 76 (38) | 35 (17.5) | 65 (32.5) | 24 (12) | 0.76 (0.67–0.84) | 0.65 (0.56–0.75) |
| | | Prompt D | 76 (38) | 43 (21.5) | 57 (28.5) | 24 (12) | 0.76 (0.67–0.84) | 0.57 (0.47–0.67) |
| | 3) Antibiotic prescription status | *Physician* | *64 (32)* | *22 (11)* | *78 (39)* | *36 (18)* | *0.64 (0.55–0.73)* | *0.78 (0.70–0.86)* |
| | | Prompt A | 51 (25.5) | 7 (3.5) | 93 (46.5) | 49 (24.5) | 0.51 (0.41–0.6) | 0.93 (0.88–0.98) |
| | | Prompt B | 44 (22) | 5 (2.5) | 95 (47.5) | 56 (28) | 0.44 (0.34–0.54) | **0.95 (0.90–0.99)** |
| | | Prompt C | 39 (19.5) | 5 (2.5) | 95 (47.5) | 61 (30.5) | 0.39 (0.30–0.49) | **0.95 (0.90–0.99)** |
| | | Prompt D | 41 (20.5) | 5 (2.5) | 95 (47.5) | 59 (29.5) | 0.41 (0.32–0.51) | **0.95 (0.90–0.99)** |

For each task and metric, the best performing scores are highlighted in bold.

[a]95% confidence intervals (CIs) calculated by bootstrapping.

*Physicians were provided the same prompt text as in Prompt A.

three tasks, suggesting it is challenging, but not impossible, to make accurate clinical recommendations based on the available clinical text. Second, while the actual outcomes of ED patients were used as the ground-truth labels in this study, it is possible that this may not reflect the best-practice care which patients should have received, nor does it reflect a standard of care that would be given across institutions. Given the nuances of real-world clinical decision-making and the variation in clinical practice across countries, it is imperative that LLMs are evaluated across different settings to ensure representative performance. Third, we only trialled three iterations of prompt engineering, in addition to our initial prompt, and this was done in a zero-shot manner. Further attempts to refine the provided prompt, or incorporate few-shot examples for in-context learning, may improve model performance[13,27–29]. Last, an evaluation of the performance of other natural language processing models, such as a fine-tuned BioClinicalBERT model or bag-of-word-based and other simpler techniques, has not been performed[30]. It is possible that these more traditional NLP models, which are typically trained or fine-tuned on a large training set of data, may outperform the zero-shot performance of GPT-like LLMs[22].

## Methods

The UCSF Information Commons contains de-identified structured clinical data as well as de-identified clinical text notes, de-identified and externally certified as previously described[31]. The UCSF Institutional Review Board determined that this use of the de-identified data within the UCSF Information Commons environment is not human participants' research and, therefore, was exempt from further approval and informed consent.

We identified all adult visits to the University of California San Francisco (UCSF) Emergency Department (ED) from 2012 to 2023 with an ED Physician note present within Information Commons (Fig. 1). Regular expressions were used to extract the *Presenting History* (consisting of 'Chief Complaint', 'History of Presenting Illness' and 'Review of Systems') and *Physical Examination* sections from each note (Supplementary Information).

**Table 3 | Comparison of physician performance with (a) GPT-3.5-turbo performance and (b) GPT-4-turbo performance across four iterations of prompt engineering [Prompt A-D] evaluated on an unbalanced n = 1000 sample reflective of the real-world distribution of clinical recommendations among patients presenting to ED, for the following three clinical recommendation tasks: (1) should the patient be admitted to hospital; (2) does the patient require radiological investigation; and (3) does the patient require antibiotics**

|  | Task |  | True positives, n (%) | False positives, n (%) | True negatives, n (%) | False Negatives, n (%) | Sensitivity (95% CI) | Specificity (95% CIa) | Accuracy (95% CIa) |
|---|---|---|---|---|---|---|---|---|---|
| a) GPT-3.5-turbo | 1) Admission status | Physician | 151 (15.1) | 79 (7.9) | 683 (68.3) | 87 (8.7) | 0.63 (0.57–0.69) | **0.90 (0.88–0.92)** | **0.83 (0.81–0.86)** |
|  |  | Prompt A | 237 (23.7) | 714 (71.4) | 48 (4.8) | 1 (0.1) | **1.00 (0.99–1.00)** | 0.06 (0.05–0.08) | 0.29 (0.26–0.31) |
|  |  | Prompt B | 234 (23.4) | 514 (51.4) | 248 (24.8) | 4 (0.4) | 0.98 (0.97–1) | 0.33 (0.29–0.36) | 0.48 (0.45–0.51) |
|  |  | Prompt C | 232 (23.2) | 475 (47.5) | 287 (28.7) | 6 (0.6) | 0.98 (0.95–0.99) | 0.38 (0.34–0.41) | 0.52 (0.49–0.55) |
|  |  | Prompt D | 226 (22.6) | 463 (46.3) | 299 (29.9) | 12 (1.2) | 0.95 (0.92–0.98) | 0.39 (0.36–0.43) | 0.53 (0.50–0.56) |
|  | 2) Radiological investigation(s) request status | Physician | 527 (52.7) | 109 (10.9) | 261 (26.1) | 103 (10.3) | 0.84 (0.81–0.86) | **0.70 (0.66–0.75)** | **0.79 (0.76–0.82)** |
|  |  | Prompt A | 619 (61.9) | 314 (31.4) | 56 (5.6) | 11 (1.1) | **0.98 (0.97–0.99)** | 0.15 (0.11–0.19) | 0.67 (0.65–0.70) |
|  |  | Prompt B | 604 (60.4) | 274 (27.4) | 96 (9.6) | 26 (2.6) | 0.96 (0.94–0.97) | 0.26 (0.21–0.31) | 0.70 (0.67–0.73) |
|  |  | Prompt C | 604 (60.4) | 268 (26.8) | 102 (10.2) | 26 (2.6) | 0.96 (0.94–0.97) | 0.28 (0.23–0.32) | 0.71 (0.68–0.74) |
|  |  | Prompt D | 608 (60.8) | 276 (27.6) | 94 (9.4) | 22 (2.2) | 0.97 (0.95–0.98) | 0.25 (0.21–0.30) | 0.70 (0.67–0.73) |
|  | 3) Antibiotic prescription status | Physician | 96 (9.6) | 142 (14.2) | 686 (68.6) | 76 (7.6) | 0.56 (0.48–0.63) | **0.83 (0.80–0.85)** | **0.78 (0.76–0.81)** |
|  |  | Prompt A | 162 (16.2) | 642 (64.2) | 186 (18.6) | 10 (1.0) | **0.94 (0.90–0.97)** | 0.22 (0.20–0.25) | 0.35 (0.32–0.38) |
|  |  | Prompt B | 159 (15.9) | 594 (59.4) | 234 (23.4) | 13 (1.3) | 0.92 (0.88–0.96) | 0.28 (0.25–0.31) | 0.39 (0.36–0.42) |
|  |  | Prompt C | 158 (15.8) | 596 (59.6) | 232 (23.2) | 14 (1.4) | 0.92 (0.87–0.96) | 0.28 (0.25–0.31) | 0.39 (0.36–0.42) |
|  |  | Prompt D | 155 (15.5) | 552 (55.2) | 276 (27.6) | 17 (1.7) | 0.90 (0.85–0.94) | 0.33 (0.30–0.37) | 0.43 (0.40–0.46) |
| b) GPT-4-turbo | 1) Admission status | Physician | 151 (15.1) | 79 (7.9) | 683 (68.3) | 87 (8.7) | 0.63 (0.57–0.69) | **0.90 (0.88–0.92)** | **0.83 (0.81–0.86)** |
|  |  | Prompt A | 237 (23.7) | 568 (56.8) | 194 (19.4) | 1 (0.1) | **1.00 (0.99–1.00)** | 0.25 (0.23–0.29) | 0.43 (0.40–0.46) |
|  |  | Prompt B | 233 (23.3) | 447 (44.7) | 315 (31.5) | 5 (0.5) | 0.98 (0.96–1.00) | 0.41 (0.38–0.45) | 0.55 (0.52–0.58) |
|  |  | Prompt C | 232 (23.2) | 417 (41.7) | 345 (34.5) | 6 (0.6) | 0.98 (0.95–1.00) | 0.45 (0.42–0.49) | 0.58 (0.55–0.61) |
|  |  | Prompt D | 233 (23.3) | 460 (46.0) | 302 (30.2) | 5 (0.5) | 0.98 (0.96–1.00) | 0.40 (0.36–0.43) | 0.54 (0.50–0.57) |
|  | 2) Radiological investigation(s) request status | Physician | 527 (52.7) | 109 (10.9) | 261 (26.1) | 103 (10.3) | 0.84 (0.81–0.86) | **0.70 (0.66–0.75)** | **0.79 (0.76–0.82)** |
|  |  | Prompt A | 562 (56.2) | 188 (18.8) | 182 (18.2) | 68 (6.8) | **0.89 (0.87–0.92)** | 0.49 (0.44–0.54) | 0.74 (0.72–0.77) |
|  |  | Prompt B | 503 (50.3) | 128 (12.8) | 242 (24.2) | 127 (12.7) | 0.80 (0.77–0.83) | 0.65 (0.60–0.70) | 0.74 (0.72–0.77) |
|  |  | Prompt C | 481 (48.1) | 112 (11.2) | 258 (25.8) | 149 (14.9) | 0.76 (0.73–0.80) | 0.70 (0.65–0.74) | 0.74 (0.71–0.77) |
|  |  | Prompt D | 521 (52.1) | 147 (14.7) | 223 (22.3) | 109 (10.9) | 0.83 (0.80–0.85) | 0.60 (0.55–0.65) | 0.74 (0.72–0.77) |
|  | 3) Antibiotic prescription status | Physician | 96 (9.6) | 142 (14.2) | 686 (68.6) | 76 (7.6) | 0.56 (0.48–0.63) | 0.83 (0.80–0.85) | 0.78 (0.76–0.81) |
|  |  | Prompt A | 106 (10.6) | 117 (11.7) | 711 (71.1) | 66 (6.6) | **0.62 (0.54–0.69)** | 0.86 (0.84–0.88) | 0.82 (0.79–0.84) |
|  |  | Prompt B | 95 (9.5) | 92 (9.2) | 736 (73.6) | 77 (7.7) | 0.55 (0.48–0.63) | 0.89 (0.87–0.91) | **0.83 (0.81–0.86)** |
|  |  | Prompt C | 82 (8.2) | 78 (7.8) | 750 (75) | 90 (9.0) | 0.48 (0.40–0.55) | **0.91 (0.89–0.93)** | **0.83 (0.81–0.86)** |
|  |  | Prompt D | 86 (8.6) | 82 (8.2) | 746 (74.6) | 86 (8.6) | 0.50 (0.43–0.58) | 0.90 (0.88–0.92) | **0.83 (0.81–0.86)** |

For each task and metric, the best performing scores are highlighted in bold.
a95% confidence intervals (CIs) calculated by bootstrapping.

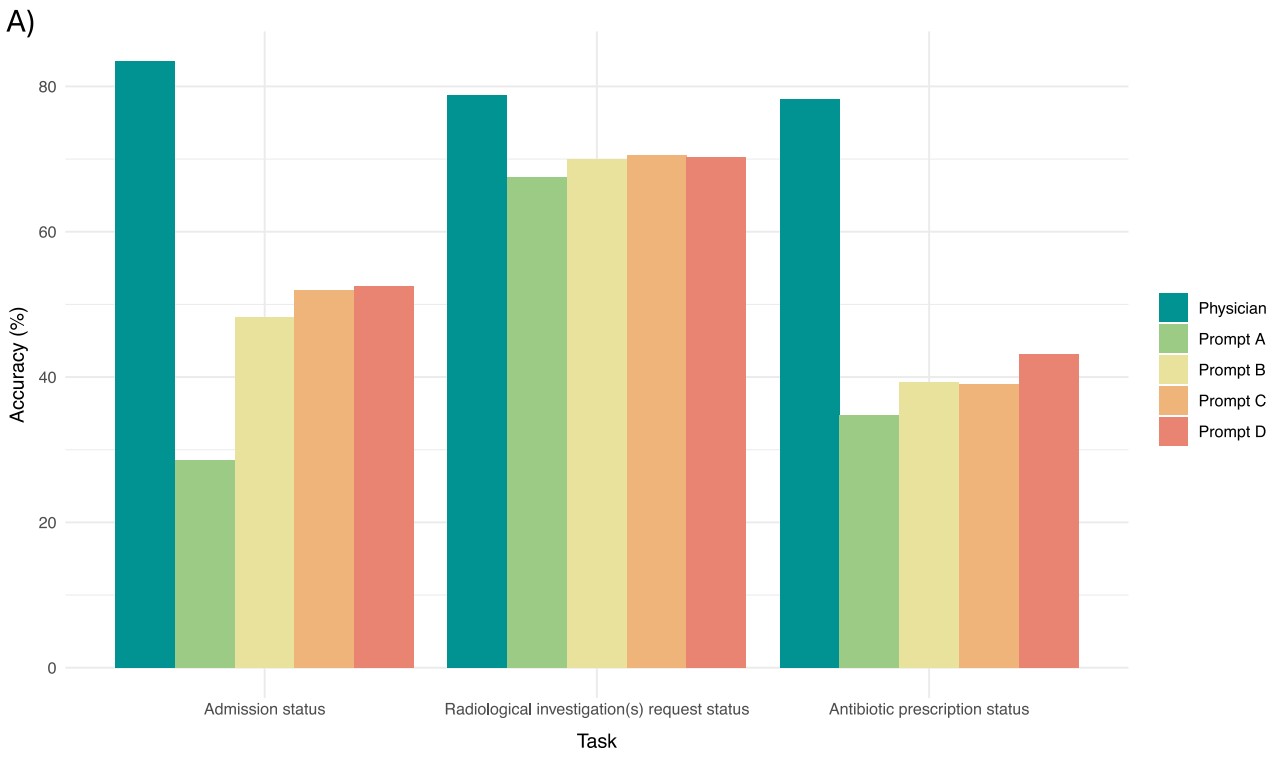

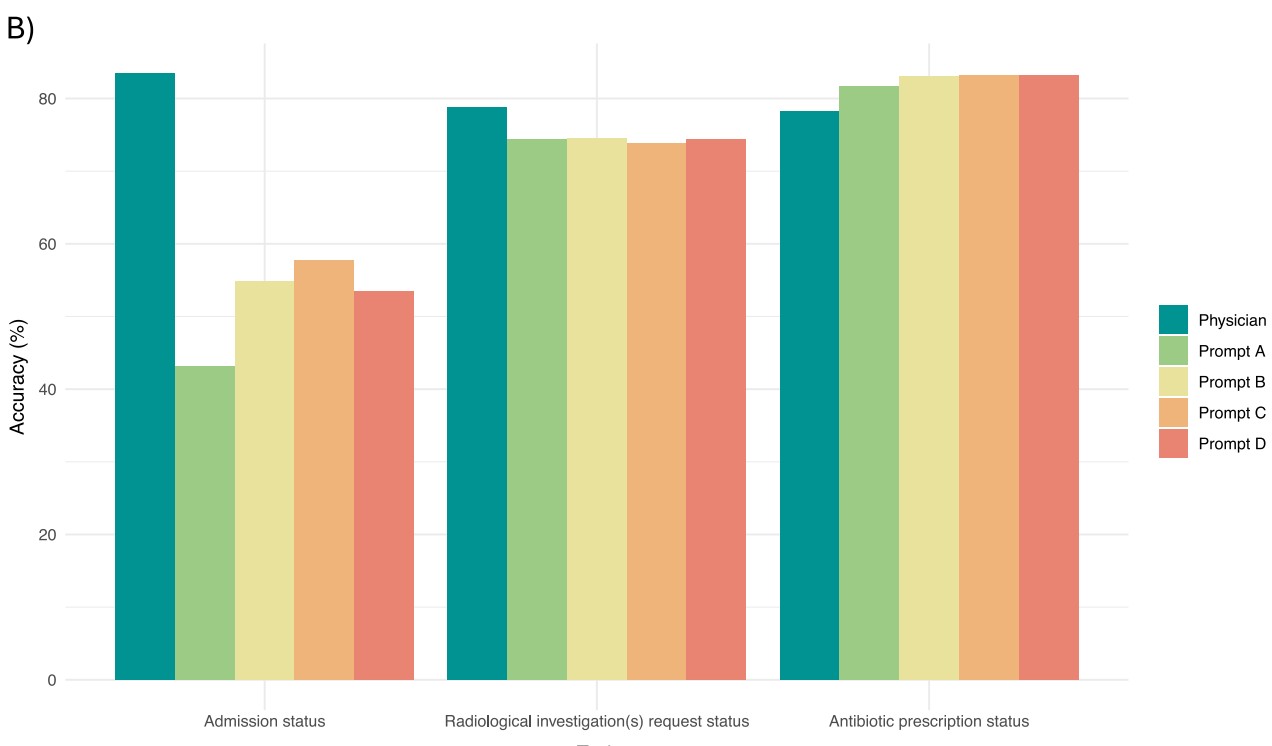

**Fig. 2 | LLM performance: unbalanced *n* = 1000 sample.** Evaluation of physician and **A** GPT-3.5-turbo or **B** GPT-4-turbo accuracy across four iterations of prompt engineering [Prompt A-D] evaluated on an unbalanced *n* = 1000 sample reflective of the real-world distribution of clinical recommendations among patients presenting to ED, for the following three clinical recommendation tasks: (1) Should the patient be admitted to hospital; (2) Does the patient require radiological investigation; and (3) Does the patient require antibiotics. Source data are provided as a Source Data file.

We sought to evaluate GPT-3.5-turbo and GPT-4-turbo performance on three binary clinical recommendation tasks, corresponding to the following outcomes: (1) *Admission status*—whether the patient should be admitted from ED to the hospital. (2) *Radiological investigation(s) request status*—whether an X-ray, US scan, CT scan, or MRI scan should be requested during the ED visit. (3) *Antibiotic prescription status*—whether antibiotics should be ordered during the ED visit.

For each of the three outcomes, we randomly selected a balanced sample of 10,000 ED visits to evaluate LLM performance (Fig. 1). Using its secure, HIPAA-compliant Application Programming Interface (API) through Microsoft Azure, we provided GPT-3.5-turbo (model = 'gpt-3.5-turbo-0301', role = 'user', temperature = 0; all other settings at default values) and GPT-4-turbo (model = 'gpt-4-turbo-128k-1106', role = 'user', temperature = 0; all other settings at default values) with only the *Presenting History* and *Physical Examination* sections of the ED Physician's note for each ED visit and queried it to determine if (1) the patient should be admitted to hospital, (2) the patient requires radiological investigation, and (3) the patient should be prescribed antibiotics. LLM performance was evaluated against the ground-truth outcome extracted from the electronic health record. Separately, a resident physician with 2 years of postgraduate general medicine training labelled a balanced $n = 200$ subsample for each of the three tasks to allow a comparison of human and machine performance. In a similar manner to the LLMs, the physician reviewer was provided with the *Presenting History* and *Physical* Examination sections of the ED Physician's note for each ED visit and asked to use their clinical judgement to decide if the patient should be admitted to the hospital, requires radiological investigation, or should be prescribed antibiotics.

We subsequently experimented with three iterations of prompt engineering (Table S1, Supplementary Information) to test if modifications to the initial prompt could improve LLM performance. Chain-of-thought (CoT) prompting is a method found to improve the ability of LLMs to perform complex reasoning by decomposing multi-step problems into a series of intermediate steps[27]. This can be done in a zero-shot manner (zero-shot-CoT), with LLMs shown to be decent zero-shot reasoners by adding a simple prompt, 'Let's think step by step' to facilitate step-by-step reasoning before answering each question[14]. Alternatively, few-shot chain-of-thought prompting can be used, with additional examples of prompt and answer pairs either manually (manual CoT) or computationally (e.g., auto-CoT) provided and concatenated with the prompt of interest[27,28]. Current understanding of the impact of zero-shot-CoT, manual CoT, and auto-CoT prompt engineering techniques applied to clinical text is limited. In this work, we sought to focus on zero-shot-CoT and investigate the effect of adding 'Let's think step by step' to the prompt on model performance.

Our initial prompt (Prompt A) simply asked the LLM to return whether the patient should be e.g., admitted to the hospital, without any additional explanation. We additionally attempted to engineer prompts to (a) reduce the high false positive rate of LLM recommendations (Prompt B) and (b) examine whether zero-shot chain-of-thought prompting could improve LLM performance (Prompts C and D). Attempting to reduce the high LLM false positive rate, Prompt B was constructed by adding an additional sentence to Prompt A: 'Only suggest *clinical recommendation* if absolutely required'. This modification was kept for Prompts C and D, which were constructed to examine chain-of-thought prompting. Because chain-of-thought prompting is most effective when the LLM provides reasoning in its output, we removed the instruction 'Please do not return any additional explanation' from Prompts C and D, and added the chain-of-thought prompt 'Let's think step by step' to Prompt D, increasing GPT-3.5-turbo but not GPT-4-turbo response verbosity (Table S2, Supplementary Information). Prompt C, therefore, served as a baseline for comparison of LLM performance when it is permitted to return additional explanation (in addition to its outcome recommendation), allowing comparisons with both Prompt A (where no additional explanations were allowed in the prompt) and Prompt D (where the effect of chain-of-thought prompting was examined).

To evaluate the performance of both LLMs in a real-world setting, we constructed a random, unbalanced sample of 1000 ED visits where the distribution of patient outcomes (i.e., admission status,

radiological investigation(s) request status, and antibiotic prescription status) mirrored the distributions of patients presenting to ED from our main cohort. The *Presenting History* and *Physical Examination* sections of the ED Physician's note for each ED visit were again passed to the API in an identical manner to the balanced datasets, while a resident physician was provided with these same sections and asked to manually label the entire sample to allow human vs machine comparison. In addition, an attending emergency medicine physician independently classified 10% of this subsample, with 79% concordance and comparable accuracy between reviewers (Table S3, Supplementary Information).

## Sensitivity analysis

Due to the stochastic nature of LLMs, it is possible that the order of labels reported in the original prompt may affect the subsequent labels returned. To test this, we conducted a sensitivity analysis on a balanced $n = 200$ subsample for each outcome where the positive outcome was referenced before the negative outcome in the initial prompt (e.g., '1: Patient should be admitted to hospital' precedes '0: Patient should not be admitted to hospital' in the GPT-3.5-turbo prompt).

## Statistical analysis

To assess model performance for the unbalanced datasets, the following evaluation metrics were calculated: true positive rate, true negative rate, false positive rate, false negative rate, sensitivity and specificity. Classification accuracy was calculated in addition to the aforementioned evaluation metrics utilised for the balanced datasets to provide a summative evaluation metric for this real-world simulated task. 95% confidence intervals were calculated by bootstrapping 1000 times, with replacement. All analyses were conducted in Python, version 3.11.

## Reporting summary

Further information on research design is available in the Nature Portfolio Reporting Summary linked to this article.

# Data availability

All data supporting the findings described in this manuscript are available in the article, Supplementary Information, or from the corresponding author upon request. The UCSF Information Commons database is available to individuals affiliated with UCSF who can contact the UCSF's Clinical and Translational Science Institute (CTSI) (ctsi@ucsf.edu) or the UCSF's Information Commons team for more information (info.commons@ucsf.edu). If the reader is not affiliated with UCSF, they can contact Atul Butte (atul.butte@ucsf.edu) to discuss official collaboration. Requests should be processed within a couple of weeks. Source data are provided with this paper.

# Code availability

The code accompanying this manuscript is available at https://github.com/cykwilliams/GPT-3.5-Clinical-Recommendations-in-Emergency-Department/[32].

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

## Acknowledgements

Dr Aaron E. Kornblith is supported by Eunice Kennedy Shriver National Institute of Child Health and Human Development of the National Institutes of Health under award number K23HD110716. The authors acknowledge the use of the UCSF Information Commons computational research platform, developed and supported by UCSF Bakar Computational Health Sciences Institute. The authors also thank the UCSF AI Tiger Team, Academic Research Services, Research Information Technology, and the Chancellor's Task Force for Generative AI for their software development, analytical and technical support related to the use of Versa API gateway (the UCSF secure implementation of LLMs and generative AI via API gateway), Versa chat (the chat user interface), and related data asset and services.

## Author contributions

Concept and design: Williams, Miao. Acquisition, analysis, or interpretation of data: Williams, Kornblith, Miao. Drafting of the manuscript: Williams. Critical review of the manuscript for important intellectual content: All authors. Statistical analysis: Williams, Miao. Supervision: Williams, Butte.

## Competing interests

C.Y.K.W. has no conflicts of interest to disclose. B.Y.M. is an employee of SandboxAQ. A.E.K. is a co-founder and consultant to CaptureDx. A.J.B. is a co-founder and consultant to Personalis and NuMedii; consultant to Mango Tree Corporation, and in the recent past, Samsung, 10x Genomics, Helix, Pathway Genomics, and Verinata (Illumina); has served on paid advisory panels or boards for Geisinger Health, Regenstrief Institute, Gerson Lehman Group, AlphaSights, Covance, Novartis, Genentech, and Merck, and Roche; is a shareholder in Personalis and NuMedii; is a minor shareholder in Apple, Meta (Facebook), Alphabet (Google), Microsoft, Amazon, Snap, 10x Genomics, Illumina, Regeneron, Sanofi, Pfizer, Royalty Pharma, Moderna, Sutro, Doximity, BioNtech, Invitae, Pacific Biosciences, Editas Medicine, Nuna Health, Assay Depot, and Vet24seven, and several other non-health related companies and mutual funds; and has received honoraria and travel reimbursement for invited talks from Johnson and Johnson, Roche, Genentech, Pfizer, Merck, Lilly, Takeda, Varian, Mars, Siemens, Optum, Abbott, Celgene, AstraZeneca, AbbVie, Westat, and many academic institutions, medical or disease specific foundations and associations, and health systems. A.J.B. receives royalty payments through Stanford University, for several patents and other disclosures licensed to NuMedii and Personalis. A.J.B.'s research has been funded by NIH, Peraton (as the prime on an NIH contract), Genentech, Johnson and Johnson, FDA, Robert Wood Johnson Foundation, Leon Lowenstein Foundation, Intervalien Foundation, Priscilla Chan and Mark Zuckerberg, the Barbara and Gerson Bakar

Foundation, and in the recent past, the March of Dimes, Juvenile Diabetes Research Foundation, California Governor's Office of Planning and Research, California Institute for Regenerative Medicine, L'Oreal, and Progenity. None of these entities had any bearing on the design of this study or the writing of the manuscript.
