## [Peer Review File · Nature Communications]

Reviewers' Comments:

Reviewer #1:

Remarks to the Author:

Thank you for the opportunity to review this interesting paper.

ChatGPT and other AI programs are seemingly attractive for clinicians with a view that this should work and improve accuracy or speed of diagnosis, prognosis or management.

However, data to show that LLMs can actually achieve this for front line clinicians using real or at least realistic data sets are scant.

This paper assesses the accuracy of ChatGPT 3.5 in determining 3 relevant clinical questions – admission or not, radiography or not and antibiotics or not – using real patient record data abstracted for recorded history and physical examination and compares it to an independent clinician assessment related to the same information and using ultimate recorded clinical decisions as the outcome.

In my copy of the paper the methods section appeared after the results section and the acknowledgement and COI statements were duplicated.

Introduction and methods are generally adequate. However, the independent clinician reviewer/s needs much more details: was this one reviewer or many? What was their level of residency training? What was their level of training or instructions for this specific project if any?

I was initially a bit confused how the results are presented in the text - I was looking for a comparison of sensitivity using the calculated values and measures of precision between the two groups not the sensitivity of one group vs the range of the other group. I then realised the range referred to the range of sensitivities of PROMPT A to D. Would add this in the relevant results sections to clarify. However would still add measures of precision for sens/spec in the tables.

A limitation worth discussing is the lack of gold standard practice in the patients in the data set. While it seems reasonable as a first step to focus on the actual outcome of the emergency patients at UCSF (ie was the patient admitted/radiographed/received antibiotics at UCSF) was this practice actually the one that the patients should have received or reflect care elsewhere? Who provided this care- a trainee clinician or senior staff? Also, care varies highly across countries for example and the AI generated sensitivity/specificity data may look different in another country.

A further limitation to discuss is the seniority of the human reviewer- would senior medical staff as compared to resident physicians produce a different/higher accuracy differential?

Reviewer #2:

Remarks to the Author:

It's note worthy that the base GPT-3.5 turbo model performed so poorly and this likely does not match the experience of many physicians who are surreptitiously using the current chatGPT interface. Additionally it's worth showing the improvements in output with more complex prompt engineering.

There has been similar work done re: benefits of using the GPT-3.5 model but one real question I have is - GPT-4 (and turbo) is out and what are the improvements here? we fully expect improvements with this larger and more complex model and GPT-3.5 to be phased out within the next few months.

REVIEWER COMMENTS

Reviewer #1 (Remarks to the Author):

Thank you for the opportunity to review this interesting paper.

ChatGPT and other AI programs are seemingly attractive for clinicians with a view that this should work and improve accuracy or speed of diagnosis, prognosis or management.

However, data to show that LLMs can actually achieve this for front line clinicians using real or at least realistic data sets are scant.

This paper assesses the accuracy of ChatGPT 3.5 in determining 3 relevant clinical questions – admission or not, radiography or not and antibiotics or not – using real patient record data abstracted for recorded history and physical examination and compares it to an independent clinician assessment related to the same information and using ultimate recorded clinical decisions as the outcome.

- Response: We thank the reviewer for their positive feedback.

In my copy of the paper the methods section appeared after the results section and the acknowledgement and COI statements were duplicated.

- Response: We have followed the guidance in the journal's *Instructions for Authors* here and ordered the manuscript sections: Introduction, Results, Discussion, Methods. We would be happy to rearrange the sections at the request of the editor.

Introduction and methods are generally adequate. However, the independent clinician reviewer/s needs much more details: was this one reviewer or many? What was their level of residency training? What was their level of training or instructions for this specific project if any?

- Response: We thank the reviewer for their comment. The clinician reviewer was a resident with 2 years of postgraduate general medical training. In a similar manner to the GPT model, they were prompted to read through each patients' clinical history and examination findings and return whether the patient should be admitted to hospital / requires radiological investigation / requires antibiotics using their clinical judgement. We have added these details into the Methods section of the manuscript (lines 301-307).

I was initially a bit confused how the results are presented in the text - I was looking for a comparison of sensitivity using the calculated values and measures of precision between the two groups not the sensitivity of one group vs the range of the other group. I then realised the range referred to the range of sensitivities of PROMPT A to D. Would add this in the relevant results sections to clarify.

- Response: We thank the reviewer for this feedback. We agree that this was not clear and have changed the presentation of the Results to clarify that this is a range of accuracy scores across Prompts A to D (lines 143-145).

However would still add measures of precision for sens/spec in the tables.

- Response: We have calculated bootstrapped 95% confidence intervals and added these to the Tables as requested.

A limitation worth discussing is the lack of gold standard practice in the patients in the data set. While it seems reasonable as a first step to focus on the actual outcome of the emergency patients at UCSF (ie was the patient admitted/radiographed/received antibiotics at UCSF) was this practice actually the one that the patients should have received or reflect care elsewhere? Who provided this care- a trainee clinician or senior staff? Also, care varies highly across countries for example and the AI generated sensitivity/specificity data may look different in another country.

- We thank the reviewer for these suggestions. We have included a discussion of the limitations of using actual outcomes as the ground truth labels, and the need for LLM evaluation across different settings, in our Limitations section (lines 248-254).

A further limitation to discuss is the seniority of the human reviewer- would senior medical staff as compared to resident physicians produce a different/higher accuracy differential?

- We thank the review for this comment. We have added the results of annotations by an attending Emergency Medicine physician in Table S3, demonstrating comparable accuracy between resident and attending physician reviewers.

Reviewer #2 (Remarks to the Author):

It's note worthy that the base GPT-3.5 turbo model performed so poorly and this likely does not match the experience of many physicians who are surreptitiously using the current chatGPT interface. Additionally it's worth showing the improvements in output with more complex prompt engineering.

- We thank the reviewer for their comments. We agree that it is useful to evaluate improvements in LLM performance with more complex prompt engineering, which is why we have included 4 iterations of increasingly complex prompts in this study.
- While the results of further prompt engineering may be interesting, we believe it is equally important to expediently validate and report zero-shot LLM performance and therefore consider more complex prompt engineering strategies to be outside the scope of this study.

There has been similar work done re: benefits of using the GPT-3.5 model but one real question I have is - GPT-4 (and turbo) is out and what are the improvements here? we fully expect improvements with this larger and more complex model and GPT-3.5 to be phased out within the next few months.

- We thank the reviewer for their comments. We have repeated the analysis with GPT-4-turbo, which had not been previously available at our institution.
- We found that GPT-4 performance was greater than that of its predecessor, but on average remained inferior to physician, with a similar tendency to excessively recommend intervention, leading to a high false positive rate. We have updated the Results and Discussion to reflect these additional analyses.

Reviewers' Comments:

Reviewer #1:

Remarks to the Author:

Thanks for addressing my comments.

It would have been helpful to either spell out in detail in the response letter where exactly changes were made or highlight or track changes in the revised manuscript.

Reviewer #2:

None